# Equivalence and endothelial safety of temporary intracoronary shunt device: Insights from preclinical porcine and rabbit models

Qi Chen[1], Yinfen Wang[1], Liuhuan Huang[2]*, Bin Li[3], Zhuo Chen[4], Ping Liu[4], Xin Jiang[4]*

**1** Department of rehabilitation medicine, The Eighth Affiliated Hospital, Sun Yat-sen University, Shenzhen, China, **2** Department of Extracorporeal Circulation, First People's Hospital of Foshan, Foshan, China, **3** Beijing Key Laboratory of Pre-clinical Research and Evaluation for Cardiovascular Implant Materials, Animal Experimental Centre, Fuwai Hospital, National Centre for Cardiovascular Disease, Chinese Academy of Medical Sciences and Peking Union Medical College, Beijing, China, **4** PerMedos Medical Technology Co., LTD., Shanghai, China

\* jiangxin@cardimed.com.cn (XJ); 296833197@qq.com (LHH)

## Abstract

### Background

This preclinical investigation evaluated the operational equivalence of a temporary intracoronary shunt (TICS) device while documenting patterns of endothelial repair during the perioperative period.

### Methods

Porcine coronary bypass models were established using off-pump (OPCAB) and on-pump beating-heart surgical approaches. Through a triple randomization design, test devices and predicate counterparts were deployed in left/right coronary arteries. Equivalence was determined at postoperative day 7 through tripartite analysis: 1) sequential laboratory measurements (preoperative to 7-day follow-up), 2) histopathological evaluation of vascular specimens, and 3) angiographic assessment. Extreme-sized device validation employed rabbit carotid arteries (small-caliber vascular bed) and abdominal aortae (large-caliber model), with equivalence criteria encompassing hemodynamic stability (flow velocity, trans-device pressure differentials) and microarchitectural preservation (endothelial integrity, internal elastic lamina continuity).

### Results

In the conventional device cohort (12 target vessels), no perioperative type 5 myocardial infarction occurred. Postoperative CTA confirmed patent lumens and unobstructed distal flow in all vessels. Cardiac biomarkers (troponin, CK-MB, myoglobin) showed no significant differences at preoperative, 4h, 8h, 24h, 72h, or 7-day timepoints (P ≥ 0.05). For extreme-sized devices, hemodynamic parameters (mean proximal/distal pressure: test vs. predicate, P ≥ 0.05) and endothelial outcomes, including

**Data availability statement:** All relevant data are within the manuscript and its Supporting Information files.

**Funding:** This work was supported by PerMedos Medical Technology Co., LTD and the Shenzhen Futian District Health System Research Project (FTWS2022037). PerMedos provided financial support and participated in study design through its employees (Xin Jiang, Zhuo Chen and Ping Liu) but had no role in data collection, analysis, or interpretation, and the Shenzhen Futian project supported data collection and analysis. Neither funder was involved in data interpretation or manuscript preparation.

**Competing interests:** The authors have declared that no competing interests exist.

elastic lamina injury scores, demonstrated equivalence between test and predicate devices.

## Conclusion

Functional parity between TICS and predicate devices was established in both porcine and lagomorph models, with observed endothelial alterations demonstrating transient characteristics limited to the acute perioperative window.

## Introduction

Coronary artery bypass grafting (CABG) maintains its position as the gold-standard surgical intervention for advanced coronary artery disease [1]. The refined off-pump CABG (OPCABG) technique, which eliminates cardiopulmonary bypass requirements, has gained substantial clinical traction as an effective alternative to conventional approaches. Clinical evidence confirms this modification significantly attenuates systemic inflammatory activation while reducing perioperative morbidity, hospitalization duration, and procedural costs advantages that have secured its integration into modern cardiac surgical practice [2].

A major technical hurdle during OPCABG anastomosis involves maintaining precise vessel stabilization while establishing a hemorrhage-controlled operative field. Temporary intracoronary shunts (TICS) address this challenge through their unique hemodynamic design: by channeling proximal coronary flow through their luminal architecture, these devices simultaneously stabilize the anastomotic site and preserve distal myocardial perfusion. Following Trapp's pioneering demonstration of their utility in beating-heart revascularization [3], TICS devices have evolved into essential instrumentation for OPCABG procedures, effectively maintaining coronary flow continuity, minimizing ischemic burden, and preventing contractile impairment during distal anastomosis construction.

According to the Technical Guideline for Clinical Evaluation Report for Registration Application of Medical Devices (NMPA 2021), TICS may be exempt from human clinical trials. Nevertheless, persistent procedural risks including intimal dissection and distal embolization during device deployment [4] underscore the imperative for exhaustive preclinical safety characterization, necessitating rigorous preclinical safety evaluations.Existing equivalence studies of TICS devices have primarily concentrated on hemodynamic performance metrics, leaving critical gaps in the systematic evaluation of endothelial interface interactions.

The present study aim to conduct a systematic comparative analysis of the differences between two TICS devices, includng PerMedos and Medtronic devices, during surgical procedures from multiple critical perspectives, with co-primary endpoints encompassing (1) hemodynamic stabilization efficacy and (2) vascular endothelial injury biomarkers. We hypothesized that the two devices exhibit substantial equivalence during surgery, and our findings revealed comparable intraoperative hemodynamic profiles during surgery and endothelial response patterns between PerMedos

and Medtronic devices, providing evidence for the safety interchangeability of TICS devices in OPCABG. This study bridges the critical gap by integrating equivalence evaluation with a systematic analysis of endothelial safety, thereby advancing the preclinical assessment framework for TICS.

## Materials and methods

### Study design

The investigation was conducted from July 2023 to June 2024. Six female domestic porcine (4−6 months old, body weight 40−50 kg) were supplied by Jiangsu Yadong Laboratory Animal Research Institute Co., Ltd. [Porcine Production License: SCXK (Su) 2021−0015]. Ten New Zealand White rabbits (8 weeks old, body weight 2.9–3.3 kg; sex unrestricted) were procured from Beijing Long'an Laboratory Animal Breeding Center [Rabbit Production License: SCXK (Jing) 2021−0014].

The study employed a minimally required sample size, statistically validated for power, in full compliance with China's medical device regulations and Institutional Animal Care and Use Committee (IACUC) approval [5,6]. This methodology ensured rigorous adherence to the 3R principles (Replacement, Reduction, and Refinement), maintaining scientific integrity while upholding the highest ethical standards for experimental animal welfare [7]. All experimental protocols were implemented under the supervision and approval of the Institutional Animal Care and Use Committee (IACUC), strictly adhering to the Guidelines for the Ethical Review of Laboratory Animal Welfare. The ethical approval identifiers were designated as IACUC-C2024-0001–07 (porcine studies) and BSYXIA003 (lagomorph studies). The study was approved by the Experimental Animal Welfare and Ethics Committee of China.

### Experimental facilities

Routine procedures were conducted at the surgical facility of DOCTOR MEDITECH GUANGZHOU CO.,LTD. [Laboratory Animal Use Certificate: SYXK (Yue) 2023−0316]. Extreme condition testing was performed at the surgical suite of Beijing Tonghe Litai Biotechnology Co., Ltd. [Laboratory Animal Use Certificate: SYXK (Jing) 2019−0016].All facilities maintained operational compliance with the Regulations on the Administration of Laboratory Animals (China) and National Standard (Environmental Conditions and Facilities for Laboratory Animals). Animal drinking water met the sanitary specifications defined in National Standard (Sanitary Standard for Drinking Water).

### Temporary intracoronary shunt device

Conceived and fabricated by PerMedos Medical Technology Co., LTD(Shanghai, China), consists of five key components: (1) a distal tip fabricated from radiopaque Pebax® 7233 medical-grade polymer containing 20% w/w barium sulfate additive for enhanced fluoroscopic visualization; (2) a shunt body composed of unmodified Pebax® 7233 elastomer compliant with ISO 10993 biocompatibility standards; (3) a spring mechanism constructed with ASTM F138-certified 316L surgical-grade stainless steel ensuring mechanical resilience; (4) a retrieval suture utilizing non-absorbable USP 5−0 polypropylene monofilament for controlled device extraction; and (5) a proximal marker formed from thermoplastic polyurethane (TPU) composite incorporating 20% w/w barium sulfate radiopacifier for procedural localization. Terminal sterilization was performed through ISO 11135:2014-validated ethylene oxide processing, with residual ethylene oxide and ethylene chlorohydrin concentrations maintained below 10 μg/g in compliance with ISO 10993−7 toxicological safety thresholds.(Fig 1).

### Animal preparation

To ensure anatomical and pathological similarity with humans and haemodynamic similarity, we utilized Chinese hybrid Landrace porcine models. As per the Guiding Principles for Technical Review of Animal Experimental Studies of Medical Devices from the Medical Device Technical Review Center of the National Medical Products Administration of the People's Republic of China, 6 porcine were used for this study. The animals were individually marked with ear tags for identifcation.

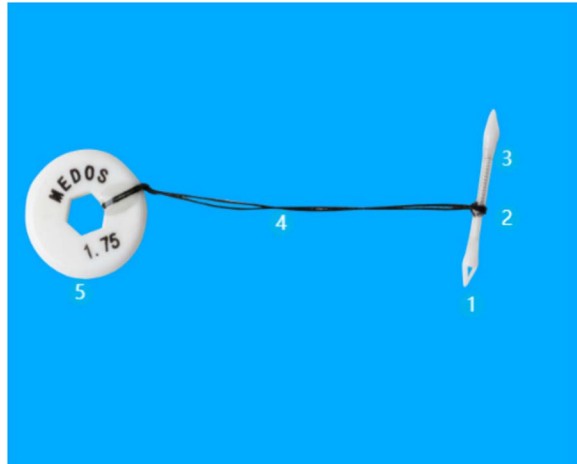

**Fig 1. Structural composition.** 1. Distal tip 2. Shunt body 3. Spring mechanism 4. Retrieval suture 5. Pull ring.

The experimental facilities and the animals were provided by an animal experiment institute (DOCTOR MEDITECH GUANGZHOU CO.,LTD, Guangzhou, China) accredited by the Standardization Administration of China for Laboratory Animals-Requirements of Environment and Housing Facilities.Prior to the operation, the animals underwent a comprehensive acclimation process in the feeding room, spanning 7–15 days, alongside a rigorous quarantine period.

## Operative procedures

**Typical specification verification.**  Six porcine subjects were stratified into two surgical groups: four underwent off-pump beating heart procedures, and two received on-pump beating heart interventions with cardiopulmonary bypass. A triple randomization protocol was implemented to govern (1) coronary target selection (left anterior descending [LAD] vs. right coronary artery [RCA]), (2) device allocation (test vs. predicate devices), and (3) procedural sequence.For off-pump interventions, systemic heparinization maintained an activated clotting time (ACT) >300 s. Median sternotomy exposed the LAD/RCA, stabilized using PerMedos Stable-P cardiac stabilizers under pharmacologically regulated mean arterial pressure (60−80 mmHg). Temporary intracoronary shunts (TICS) were deployed in randomized sequences for 5-minute intervals to simulate end-to-side anastomoses, followed by 7−0 polypropylene suture anastomosis, heparin reversal, and thoracotomy closure. On-pump procedures replicated these steps with integrated extracorporeal circulation. After the operation, the animals were transferred to the care unit for monitoring and were later transferred back to the feeding room once recovered. Each animal was kept in an individual cage and had access to food and water before and after the operation. Water was freely accessible; the feeding, which was based on in-dividual weight, was quantitative, administered twice daily at fxed times. The animal facility room temperature and relative humidity were suitable. Postoperative monitoring spanned 7 days and included:1.Continuous telemetric hemodynamic surveillance;2.Serial quantification of cardiac biomarkers (troponin I, CK-MB);3.Terminal histopathological analysis (hematoxylin-eosin staining) to evaluate endothelial integrity.

**Extreme specification verification.**  Ten New Zealand White rabbits were randomized using a block design into test and predicate device groups, which were further stratified into minimum (n = 5) and maximum-specification (n = 5) subgroups. The validations targeted two anatomical sites: the common carotid arteries (CCA), where minimum-specification devices were implanted under microscopy.The right CCA underwent proximal/distal clamping (0.5 cm from bifurcation), while a 2-mm arteriotomy was performed on the left CCA.Abdominal aortas (AA): Maximum-specification devices were deployed in infrarenal AA segments under cross-clamp ischemia. Hemodynamic parameters were recorded

using Millar Mikro transducers (proximal/distal pressure gradients at 30-s intervals for 5 min). Aortic pressure waveforms were analyzed pre-/post-device extraction using ADInstruments PowerLab systems.Terminal procedures adhered to AVMA guidelines, with euthanasia via intravenous pentobarbital (100 mg/kg). All surgeries were conducted under isoflurane anesthesia (2–3% MAC). Procedural timelines are comprehensively outlined ([Fig 2]), Selection Protocol ([Fig 3]).

### Follow-up

**Laboratory biomarker profiling.**  Serial blood samples were collected at six timepoints: preoperatively, and 4 h, 8 h, 24 h, 72 h, and 7 d postoperatively. Cardiac injury biomarkers were quantified using chemiluminescent immunoassay: cardiac troponin (cTn) with 99th percentile reference, creatine kinase-MB (CK-MB) via mass assay and myoglobin (Myo). Hematological parameters were analyzed by veterinary hematology analyzer: red blood cell count (RBC, $\times 10^{12}$/L), white blood cell count (WBC, $\times 10^9$/L), hemoglobin (HGB, g/dL), and platelet count (PLT, $\times 10^9$/L).

**Coronary computed tomography angiography (CTA) protocol.**  Preoperative and 7-day postoperative angiographic evaluations were conducted using dual-source 64-slice CT (Philips Incisive power) with retrospective ECG-gating synchronized to 70–80% of the R-R interval. The acquired datasets were subjected to tripartite analysis via the syngo. via workstation, which included: (1) distal perfusion quantification through Hounsfield unit (HU) density ratio calculations comparing contrast-enhanced luminal areas; (2) three-dimensional coronary tree reconstruction using volume-rendering techniques with adaptive vessel tracking algorithms; and (3) quantitative stenosis assessment via QCA-CMS® software (Medis Medical Imaging Systems), employing edge-detection thresholds of 150–350 HU and minimum lumen diameter measurements with 0.1 mm resolution, while ensuring radiation exposure remained below 3.5 mSv per SCCT guidelines.

**Hemodynamic monitoring & pressure gradient analysis.**  Following deployment of minimum-spec device, acute hemodynamic parameters were acquired through:

Following vascular transection, blood efflux was collected over a 10-second interval using pre-weighed sterile gauze. Mass flow rate (g/s) was converted to volumetric flow rate (mL/s) through the following equation: $Qv = \frac{m}{t \times \rho}$, where $\rho$=1.05 kg/m³ (rabbit blood density). Simultaneous proximal/distal mean arterial pressure (MAP) measurements via pressure transducers sampled at 30-s intervals for 300 s.

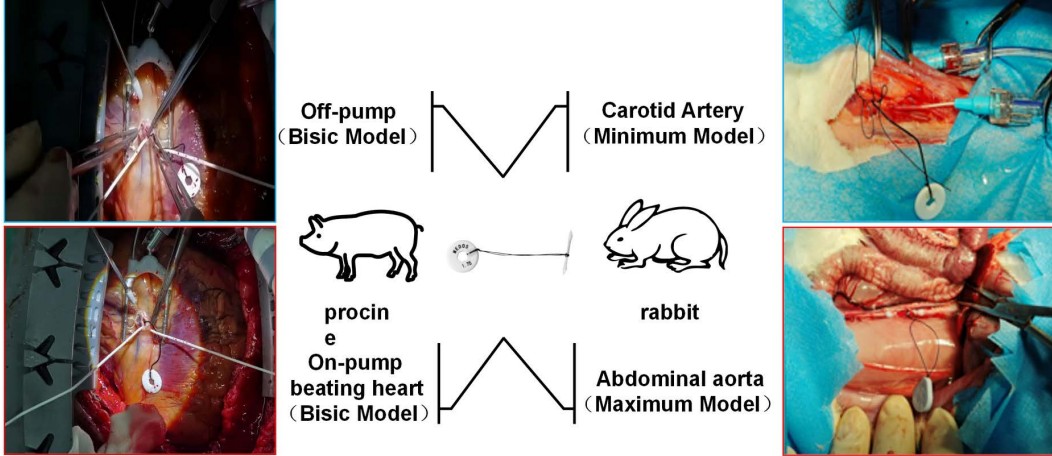

**Fig 2. Experimental workflow.** Left Panel (Upper): Off-pump beating heart procedure for left anterior descending (LAD) artery implantation.Left Panel (Lower): On-pump beating heart intervention with cardiopulmonary bypass for right coronary artery (RCA) implantation.Right Panel (Upper): Minimally invasive carotid artery implantation (common carotid artery, CCA) for hemodynamic data acquisition over a standardized 5-minute interval.Right Panel (Lower): Abdominal aortic (AA) device deployment with real-time pressure gradient monitoring (5-minute duration).

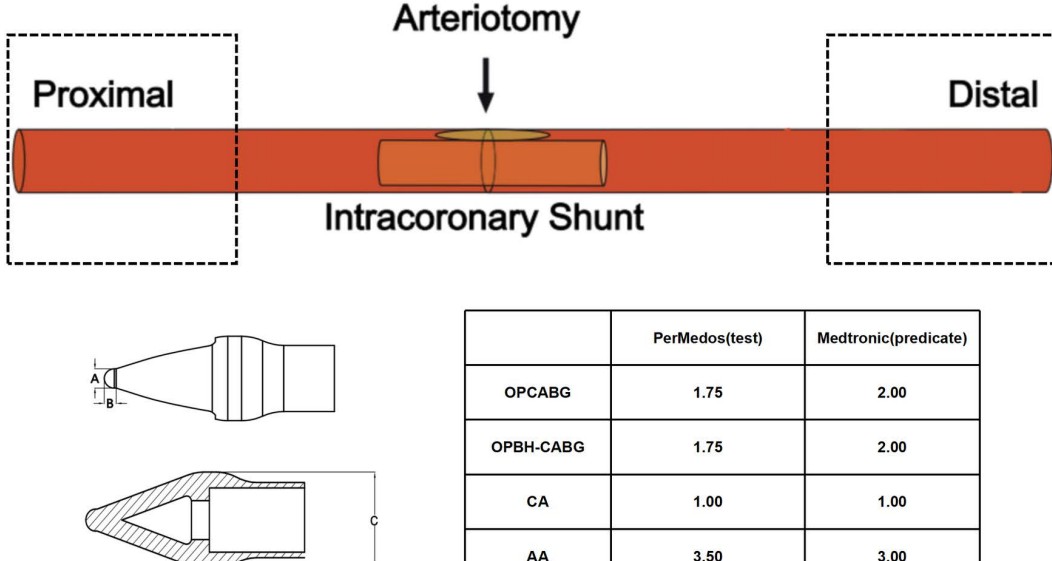

| | PerMedos(test) | Medtronic(predicate) |
|---|---|---|
| OPCABG | 1.75 | 2.00 |
| OPBH-CABG | 1.75 | 2.00 |
| CA | 1.00 | 1.00 |
| AA | 3.50 | 3.00 |

**Fig 3. Device selection protocol.** For conventional testing, test devices with nominal diameter (1.75 mm) were compared against predicate counterparts (2.00 mm; Medtronic CLEARVIEW®). Extreme specification evaluation employed: minimum-diameter cohort (1.00 mm test vs. 1.00 mm) and maximum-diameter cohort (3.50 mm test vs. 3.00 mm). All devices were implanted at the midpoint of target vessels (black arrow indication), with histopathological sampling focused on proximal and distal interface zones (dashed-line demarcation) corresponding to primary frictional contact areas between device extremities and vascular endothelium.

**Endothelial integrity and thrombogenicity assessment.** Explanted vascular specimens were systematically evaluated through a tripartite analysis that adhered to the ISO 10993–4:2017 Annex C guidelines. Gross morphological assessment was conducted using ×40 stereomicroscopy (Dako's DP260) and a validated 0–4 scoring system: 0 indicates an intact endothelium; 1 signifies endothelial denudation with an intact internal elastic lamina (IEL); 2 represents IEL disruption affecting less than 25% of the circumference; 3 indicates IEL disruption between 25–50% of the circumference; and 4 denotes a transmural injury with penetration into the medial layer.

**Histomorphometry:** Specimens were embedded in paraffin and sectioned using a Leica RM2235 microtome with a section thickness ranging from 3 to 5 micrometers. They were then stained with hematoxylin & eosin (H&E) and Masson's trichrome. All assessments were conducted by dual-blinded cardiovascular pathologists.

### Statistical analysis

All experimental data were processed using a proprietary image segmentation-based evaluation system for temporary intracoronary shunts (Chinese Patent No. 2024113858739), which incorporated automated predicate device benchmarking via Dunnett's modified t-test for multiple comparisons. Continuous variables were expressed as mean ± standard deviation (SD). Endothelial injury scores underwent normality verification through the Kolmogorov-Smirnov test (α = 0.05), followed by non-parametric analysis using the Mann-Whitney U test. Statistical significance was determined at $P < 0.05$ (two-tailed).

### Results

#### Hemodynamic analysis of extreme specification devices

In the minimum-specification cohort (carotid artery deployment), comparative analysis revealed no statistically significant differences in luminal diameter between test and control devices. Hemodynamic monitoring demonstrated equivalent

distal perfusion characteristics during the 5-minute observation window, with mean arterial pressure (MAP) gradients maintaining stability. Quantitative flow analysis revealed hemodynamic equivalence between test and control minimum-diameter devices (P = 0.615). In the maximum-specification cohort (abdominal aortas deployment), comparative analysis of test and control devices (P = 0.556) is shown (Fig 4A and 4B). Histopathological analysis of maximum-specification implants revealed comparable endothelial integrity, as depicted in the representative photomicrographs(Fig 4D and 4E). **Vascular integrity assessment:** 4D and E: Minimum-specification carotid implants exhibited preserved endothelial architecture, with intact internal elastic lamina (IEL) continuity. Maximum-specification aortic devices demonstrated focal endothelial denudation and localized IEL disruptions, as denoted by black insets and arrows in high-magnification fields (H&E staining, ×100/×400 magnification). Quantitative histomorphometric analysis revealed equivalent intimal injury scores (test: 2.3±0.7 vs. predicate: 2.5±0.6; P = 0.771) and IEL integrity indices (test: 2.1±0.9 vs. predicate: 2.0±0.8; P = 1.000).

## Comparative analysis of conventional specifications

**Hematological profiling.** Serial hematological monitoring revealed stable erythrocytic profiles throughout the observation period, with preoperative and postoperative measurements demonstrating comparable red blood cell counts, hemoglobin levels, and platelet concentrations. Notably, fluctuations in transient leukocyte levels were documented within 24–72 hours post-intervention, followed by spontaneous normalization by postoperative day 7. Importantly, cardiac biomarker trajectories remained stable across temporal measurements, maintaining consistent concentrations

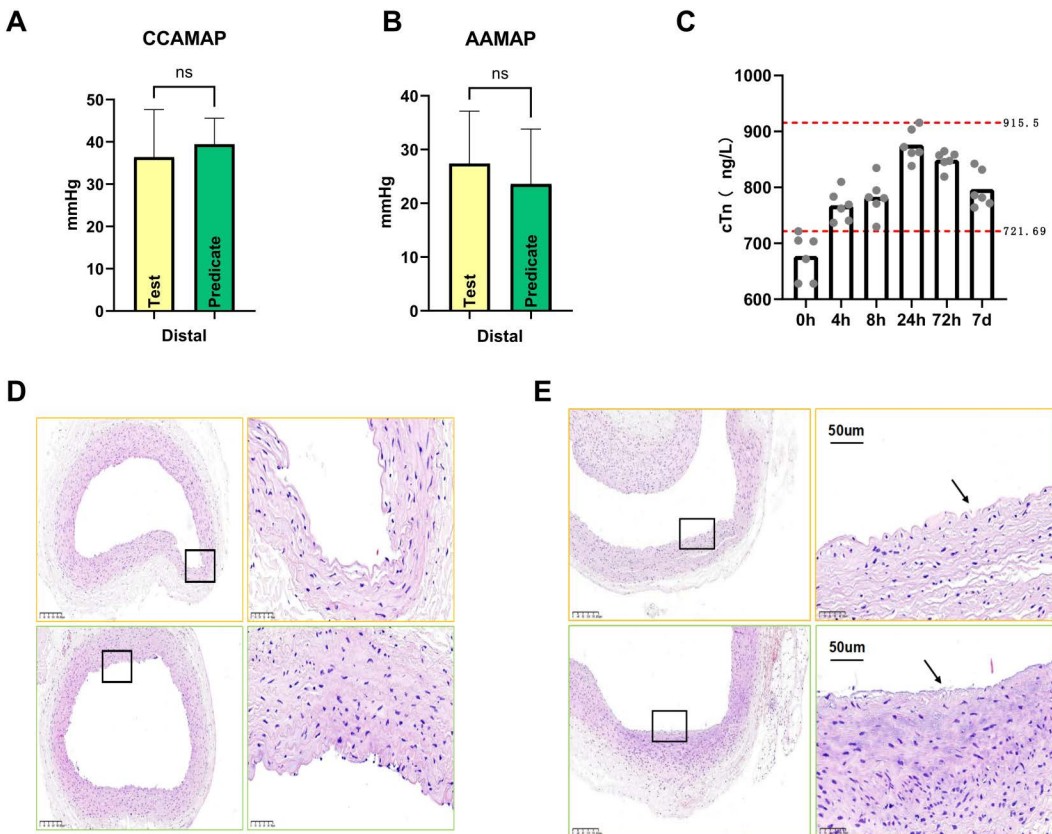

**Fig 4. Hemodynamic and histopathological analysis.** 4A and 4B: Hemodynamic Equivalence:Distal carotid artery mean arterial pressure (MAP) demonstrated comparable hemodynamic profiles between cohorts (36.4±11.22 vs 39.4±6.19 mmHg). Similarly, aortic distal MAP measurements revealed equivalent perfusion characteristics (27.4±9.74 vs 23.6±10.21 mmHg).

of cardiac troponin I, with the peak postoperative concentration (915.5 ng/L) demonstrating a 26.9% increase from the preoperative baseline (721.69 ng/L), as shown (Fig 4C).Serial measurements of creatine kinase-MB (CK-MB) isoenzyme mass concentrations and myoglobin (Myo) levels demonstrated no statistically significant perioperative variations across predefined intervals (4h, 8h, 24h, 72h, 7day postoperative vs. preoperative baseline).

**Angiographic & histopathological evaluation.** CT angiography (utilizing a dual-source 64-slice CT by Philips) revealed immediate postoperative flow in 100% of the distal vasculature, with a 7-day patency rate of 100%. The postoperative evaluation indicated that anatomical continuity at the coronary-myocardial interface was preserved, with no signs of ischemic deformation or hemodynamically significant stenosis. The anastomotic sites showed complete sealing integrity, maintaining endothelial continuity without evidence of new thrombus formation or infectious foci (Fig 5).

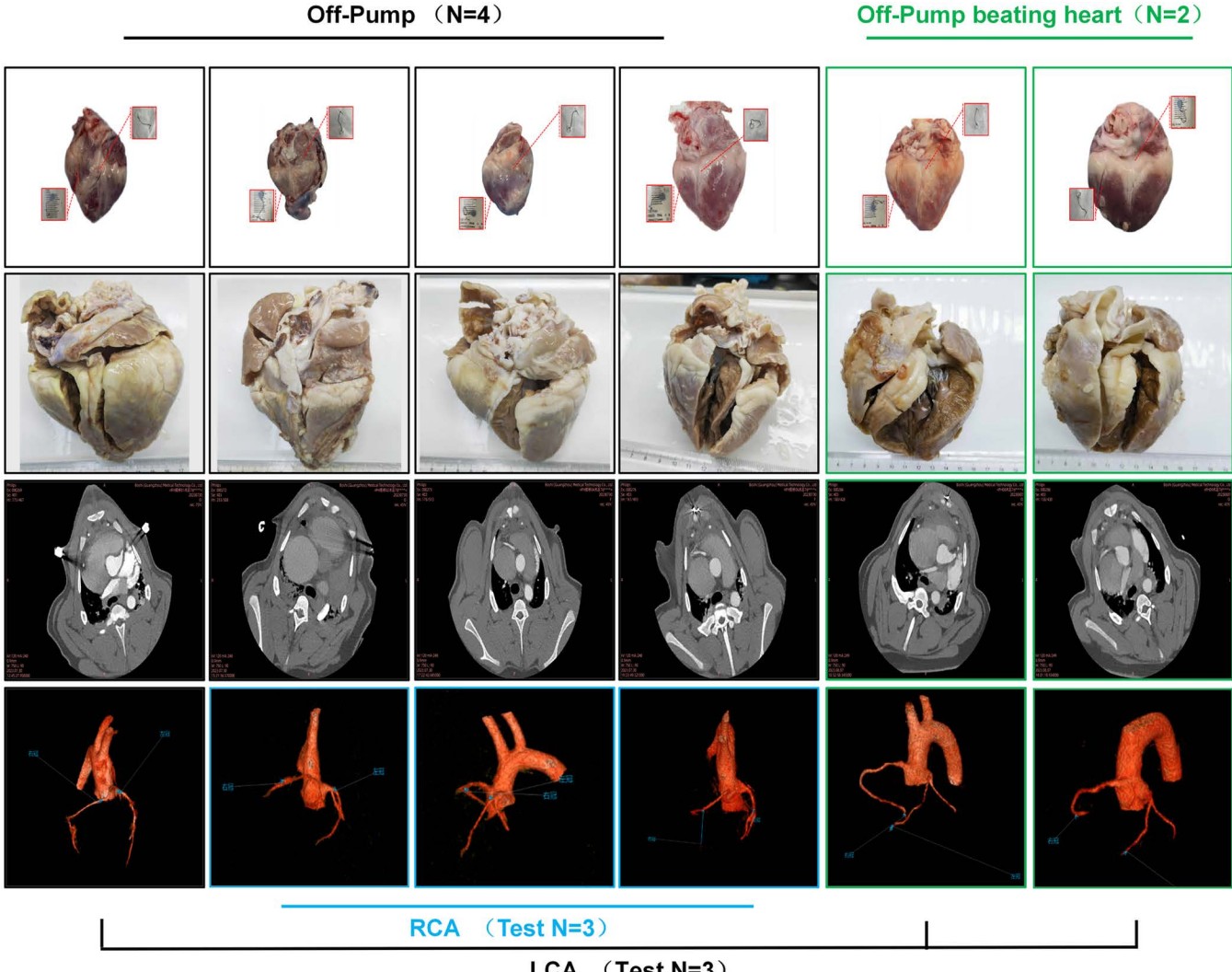

**Fig 5. Comparative surgical mapping and angiographic outcomes.** Surgical allocation analysis delineated targeted coronary interventions under distinct perfusion strategies: Off-pump beating-heart technique (n = 4, black frame) included right coronary artery deployment in 3 cases (blue tracing) and left coronary artery in 1 case (black tracing), while on-pump beating-heart procedures (n = 2, green frame) exclusively involved left coronary artery instrumentation.

**Histopathological characterization.** A comparative analysis revealed equivalent histomorphological profiles between the cohorts in the left and right anterior descending coronary arteries, demonstrating intact endothelial continuity and patent luminal architecture, as well as perivascular chronic inflammatory infiltrates accompanied by fibroproliferative responses.The myocardial cytoarchitecture remained unremarkable,showing preserved subendocardial conduction fiber alignment.Suture-line microhistology exhibited localized microhemorrhages containing organized thrombi with partial recanalization, as quantified (Fig 6).

## Discussion

Historically, regulatory approval for TICS has predominantly relied on human clinical trial data. However, contemporary frameworks now incorporate predicate device equivalence analysis, reflecting progressive refinement in regulatory science. Despite this advancement, comprehensive validation of TICS safety-efficacy profiles necessitates rigorous preclinical verification through standardized large-animal models. This involves employing quantitative biocompatibility metrics

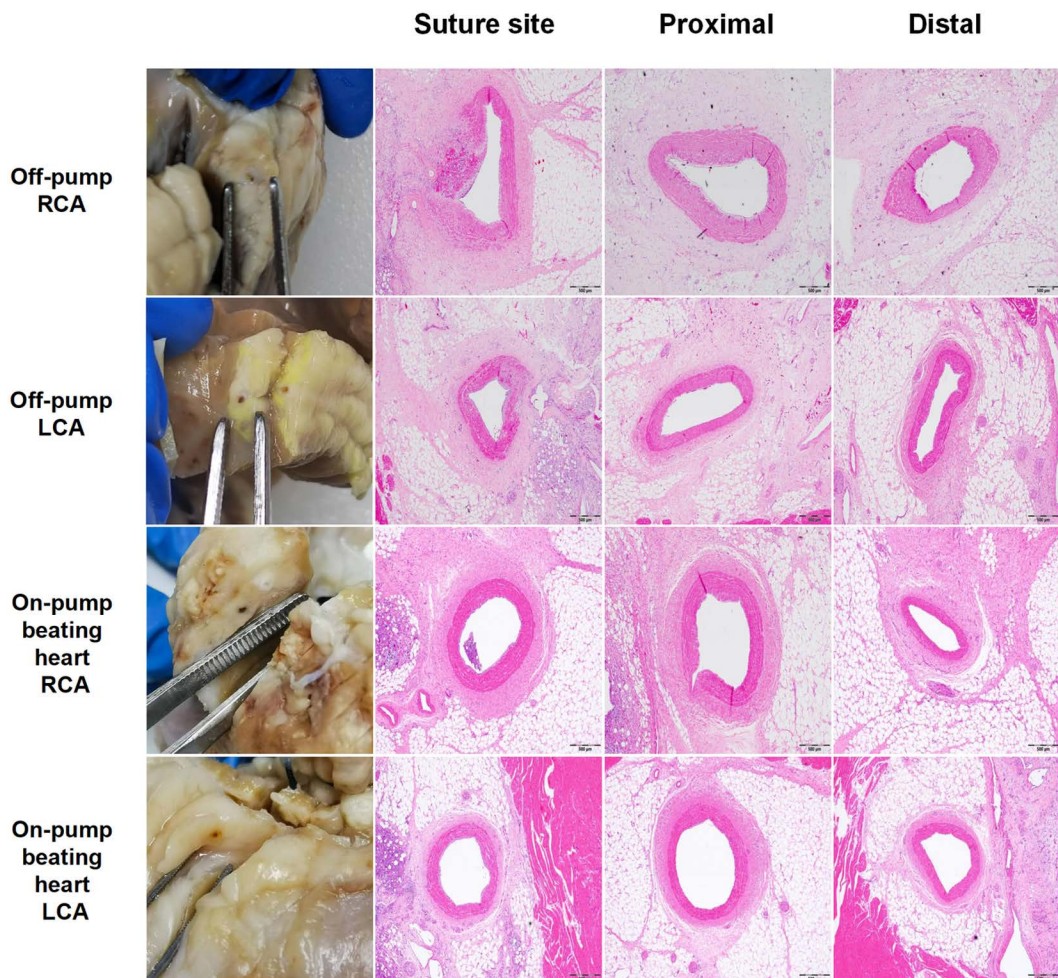

**Fig 6. Intraoperative histopathological characterization.** Under cardiopulmonary bypass, the right coronary anastomotic sites displayed localized microhemorrhagic foci (stained with H&E) that contained organized thrombi with partial recanalization, yet endothelial continuity and full luminal patency were preserved.Non-target vessels displayed preserved endothelial integrity and physiological flow parameters. Analysis of the myocardial architecture revealed intact interventricular septal cytoarchitecture and normal subendocardial conduction system morphology.

and hemodynamic stability parameters under Good Laboratory Practice (GLP) guidelines. Such methodology ensures replicable outcomes for accurate risk-benefit stratification matrices, aligning with ISO 10993 biological evaluation requirements and FDA 21 CFR 812 investigational device exemption standards. Considering the high anatomical similarity between porcine and human cardiovascular systems, this study utilized swine models to replicate human transcatheter implantation scenarios [8]. Experimental endpoints included quantitative angiographic patency indices, histomorphometric analysis of device-tissue interfaces, and hemodynamic stability monitoring to ensure a comprehensive biocompatibility assessment, aligning with ISO 10993-4:2017 biological evaluation requirements for cardiovascular devices.

The assessment of postprocedural myocardial injury, the primary diagnostic criterion for Type 5 Myocardial Infarction (MI), is defined by an elevation of cardiac troponin (cTn) exceeding 10-fold the 99th percentile upper reference limit (URL), accompanied by corroborative imaging evidence [9]. Peri-procedural myocardial injury (PMI) is classified by cTn elevations greater than 20% above baseline values [10]. In this investigation, peak preoperative cTn levels (721.69 ng/L) and 7-day postoperative values (915.5 ng/L) demonstrated a 26.9% relative increase (Δ193.81 ng/L), indicating that the myocardial injury, although present in one animal, was not of a severity that could be classified as Type 5 MI. The physiological thrombus formation observed at some suture sites of the target vessels was attributed to the lack of a continuous anticoagulation protocol and secondary prevention treatment in our simulated surgical setting [11], highlighting the importance of these measures in real-world clinical practice.

An intraprocedural analysis revealed friction-induced endothelial denudation and focal disruptions of the internal elastic lamina (IEL) during the deployment and retrieval of TICS. This necessitates strict operational protocols to reduce the risk of iatrogenic vascular injuries. Serial histopathological evaluations at the 7-day follow-up showed complete endothelial regeneration with no remaining IEL defects or neointimal hyperplasia. These results confirm the transient nature of TICS-related endothelial trauma and the inherent vascular reparative capacity, thereby supporting the safety profiles of the procedures when standardized implantation techniques are followed (shear stress <30 dyn/cm², ISO 25539–2:2020 compliance).

It is imperative to highlight that all medical devices that come into contact with the endothelium inherently pose risks of iatrogenic vascular injury. Comparative interventional cardiology studies substantiate this biomechanical continuum: shockwave balloon angioplasty induces rupture of the internal elastic lamina (IEL) with discontinuity of medial fibers [12], whereas off-pump coronary artery bypass grafting (OPCABG) triggers ischemia-mediated microvascular endothelial remodeling [13]. Notably, even in on-pump beating-heart CABG procedures, the deployment of TICS elicits measurable neointimal responses [14]. This fully confirms that vascular intimal injury is essentially an inevitable consequence of the interaction between devices and tissues.

Biomechanical simulations reveal that strategic modifications to temporary intracoronary shunt (TICS) designs significantly influence device-tissue interfacial dynamics. Miniaturized distal configurations with enhanced radial compliance demonstrate reduced endothelial friction coefficients through optimized contact surface mechanics. While these engineering refinements improve hemodynamic compatibility, acute-edge geometric profiles paradoxically predispose to hemorrhagic events via focal stress concentration phenomena, mandating meticulous control over implantation kinematics. This engineering conundrum necessitates dual-focus design philosophies where friction mitigation protocols must be counterbalanced by rigorous prevention of iatrogenic complications, particularly procedure-induced vascular dissection.

Our investigation developed a comprehensive risk evaluation framework for vascular endothelial trauma associated with temporary intracoronary shunt (TICS) deployment through comparative analysis of homologous medical devices. Importantly, real-world surgical practice demonstrates disparate clinical priorities: cardiac surgeons emphasize hemodynamic perfusion metrics and graft longevity over native vascular preservation. This divergence in clinical focus, originating from tiered therapeutic objectives where procedural efficacy hinges on immediate perfusion recovery [15], risks inadequate assessment of native vessel integrity – a critical determinant of long-term surgical success. The proposed framework reconciles this dichotomy through holistic vascular status evaluation during TICS selection and implantation,

methodically examining endothelial injury risk contributors to establish evidence-driven clinical guidance. The protocol employs standardized 1.75 mm TICS configurations that maintain protective flow parameters (40–60 ml/min) in vessels ≥1.5 mm diameter via fluid dynamic optimization, while achieving anatomical congruence Postprocedural multimodal validation – incorporating serial cardiac enzyme kinetics and coronary computed tomographic angiography (CTA) volumetric reconstruction confirmed absence of procedure-related myocardial compromise, demonstrating robust quality assurance from intraoperative biomechanical refinement through postoperative functional verification.

This study demonstrated successful clinical evaluation by comparing the target device against a control device meeting equivalent safety and efficacy benchmarks. This methodology aligns with current medical device registration frameworks established by both FDA and NMPA, specifically addressing devices with identical intended use and comparable technological characteristics. The preclinical validation demonstrated equivalence in critical performance endpoints, suggesting these findings may hold broader regulatory relevance for similarly classified devices [16]. However, brand-specific variations in material composition or design specifications may require additional verification when considering other manufacturers' products.

## Conclusions

This investigation confirms the clinical validity of Temporary Intravascular Connector Systems (TICS) through controlled comparative analysis of analogous devices, establishing their functional reliability in graft anastomosis with superior hemodynamic perfusion characteristics. Integrated histopathological evaluation alongside serial biomarker surveillance and advanced imaging modalities demonstrated that endothelial alterations secondary to TICS deployment were temporally constrained to the perioperative phase, exhibiting no detrimental effects on long-term graft performance. Mechanistic evaluations identified focal medial lamina discontinuities as the primary injury pattern, with complete endothelial restitution observed within 7 postoperative days.Clinical optimization requires strict avoidance of improper deployment techniques that could induce medioluminal dissection, with protocol refinement potentially reducing iatrogenic dissection risks. These evidence-based findings position TICS as a viable solution for microvascular anastomosis when adhering to standardized operational guidelines.

## Supporting information

**S1 Data. Experimental raw data.**
(XLS)

## Author contributions

**Conceptualization:** Xin Jiang.

**Data curation:** Bin Li.

**Formal analysis:** Qi Chen.

**Methodology:** Liuhuan Huang, Bin Li, Zhuo Chen, Ping Liu.

**Visualization:** Yinfen Wang.

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
