## [Decision Letter · Decision Letter 0]

14 May 2025

PONE-D-25-20869Equivalence and Endothelial Safety of Temporary Intracoronary Shunt Device: Insights from Preclinical Porcine and Rabbit ModelsPLOS ONE

Dear Dr. Jiang,

Thank you for submitting your manuscript to PLOS ONE. After careful consideration, we feel that it has merit but does not fully meet PLOS ONE’s publication criteria as it currently stands. Therefore, we invite you to submit a revised version of the manuscript that addresses the points raised during the review process.

**ACADEMIC EDITOR: ** While we found the work interesting, it would benefit from revision. Therefore, I invite you to respond to the reviewer's comments and submit a revised version of your article.

We look forward to receiving your revised manuscript.

Kind regards,

Redoy Ranjan, MBBS, MRCSEd, Ch.M., MS (CV&TS), FACS

Academic Editor

PLOS ONE

Journal Requirements:

Reviewers' comments:

Reviewer's Responses to Questions

**Comments to the Author**

1. Is the manuscript technically sound, and do the data support the conclusions?

Reviewer #1: Partly

Reviewer #2: Yes

Reviewer #3: Yes

2. Has the statistical analysis been performed appropriately and rigorously? 

Reviewer #1: I Don't Know

Reviewer #2: Yes

Reviewer #3: Yes

3. Have the authors made all data underlying the findings in their manuscript fully available?

Reviewer #1: Yes

Reviewer #2: Yes

Reviewer #3: Yes

4. Is the manuscript presented in an intelligible fashion and written in standard English?

Reviewer #1: Yes

Reviewer #2: Yes

Reviewer #3: Yes

5. Review Comments to the Author

Reviewer #1: Hello

This is a good article

The topic of the article is also well chosen.

I think the quality of the article is relatively good.

I hope we will see better quality articles from the authors in the future.

Good luck.

Reviewer #2: I read this paper with great interest because I used to do open chest research on dogs (40 years ago).

The surgical technique is GREAT. All the conditions to do surgical research on animals were described and adhered rigorously.

All documentations about hemodynamics and pathological specimens are GREAT. I could review the slides of the specimens of the coronary arteries

This paper is concise, with great documentation. The ENglish language is perfect

Reviewer #3: The authors present an interesting experimental study that evaluated the temporary intracoronary shunt (TICS) device while documenting patterns of endothelial repair and damage during the perioperative period. The manuscript is interesting, and has a clinically relevant question. It handles an unplorex research question in a very interesting and innovative way. The results are quite interesting.I enjoyed reading the manuscript.

I only have a few comments:

How was the sample size calculated? Does this study have enough power to suport the conclusions? Please describe within the manuscript.

The word "procine" is written wrongly troughtout the manuscript, please replace with "porcine"

What is the authors hypothesis? Please add a hypothesis to Introduction.

Can the results from this study be extrapolated to shunts from other brands? Please discuss.

6. PLOS authors have the option to publish the peer review history of their article (what does this mean? ). If published, this will include your full peer review and any attached files.

**Do you want your identity to be public for this peer review?** For information about this choice, including consent withdrawal, please see our Privacy Policy .

Reviewer #1: No

Reviewer #2: **Yes: ** Thach Nguyen MD FACC FSCAI

Reviewer #3: **Yes: ** Mateo Marin-Cuartas

---

## [Author Response · Author response to Decision Letter 1]

30 Jun 2025

Response to Reviewers

Reviewer #1: Hello

This is a good article

The topic of the article is also well chosen.

I think the quality of the article is relatively good.

I hope we will see better quality articles from the authors in the future.

Good luck.

Response We deeply appreciate your encouraging feedback. Your recognition of our work's quality is immensely uplifting, fueling our drive to strive for even greater heights in current and future endeavors. Thank you for sharing your valuable time and insightful comments.

Reviewer #2: I read this paper with great interest because I used to do open chest research on dogs (40 years ago). The surgical technique is GREAT. All the conditions to do surgical research on animals were described and adhered rigorously. All documentations about hemodynamics and pathological specimens are GREAT. I could review the slides of the specimens of the coronary arteries. This paper is concise, with great documentation. The ENglish language is perfect.

Response We are profoundly honored to receive your thoughtful review, especially given your extensive expertise in this field. Your personal experience with open-chest canine research lends exceptional weight to your feedback, and we deeply value your recognition of our surgical techniques, steadfast adherence to ethical standards, and comprehensive documentation, including hemodynamic and pathological analyses. We are particularly grateful you took the time to meticulously review the coronary artery specimens, and your validation of these findings is invaluable to our team. Your praise for the paper’s conciseness and linguistic precision is equally encouraging. Thank you for your enthusiasm and for sharing your unique perspective as a pioneer in this area. It has been a tremendous privilege to have our work evaluated by a reviewer with such profound knowledge.

Reviewer #3: The authors present an interesting experimental study that evaluated the temporary intracoronary shunt (TICS) device while documenting patterns of endothelial repair and damage during the perioperative period. The manuscript is interesting, and has a clinically relevant question. It handles an unplorex research question in a very interesting and innovative way. The results are quite interesting. I enjoyed reading the manuscript. I only have a few comments:

Comment 1: How was the sample size calculated? Does this study have enough power to support the conclusions? Please describe within the manuscript. The word "procine" is written wrongly throughout the manuscript, please replace with "porcine"

Response: We deeply appreciate the reviewer's insightful comment. The sample size calculation was conducted in strict compliance with China’s regulatory requirements, specifically adhering to the "Regulatory Consideration for Animal Study of Medical Device" issued by the Center for Medical Device Evaluation of the National Medical Products Administration of China. This guiding document establishes mandatory minimum standards for large animal utilization in medical device research, including rigorous criteria for animal species selection, sample size determination, and experimental design methodology. Simultaneously, we strictly followed the Institutional Animal Care and Use Committee (IACUC) review standards, guaranteeing all animal experiments met ethical norms and adhered to international best practices. Throughout the study, we rigorously implemented the 3R principles (Replacement, Reduction, and Refinement) as a fundamental framework. This approach enhanced the reliability and significance of our statistical data without increasing animal numbers, thereby achieving a crucial balance between scientific rigor and animal welfare in our research. We have described this content in the revised manuscript (Line 96-101).

We sincerely appreciate the reviewer’s careful reading and valuable feedback regarding the typographical errors in the original manuscript. We apologize for the typographical errors and have confirmed that all these errors have been corrected in the revised manuscript. The changes have been highlighted in the revised version for the reviewer’s convenience. Thank you for pointing this out and helping us improve the accuracy of our manuscript.

Comment 2: What is the authors hypothesis? Please add a hypothesis to Introduction.

Response Thank you for this invaluable feedback. We fully concur that explicitly stating the hypothesis will significantly enhance the clarity of our study's rationale. In the present study, we hypothesized that the two devices exhibit substantial equivalence during surgery. And our findings revealed comparable intraoperative hemodynamic profiles during surgery and endothelial response patterns between PerMedos and Medtronic devices, providing evidence for the safety interchangeability of TICS devices in OPCABG. In the revised manuscript, we have incorporated the hypothesis into the Introduction section, and we are very grateful for the reviewer’s comment that has significantly enhanced the quality of our manuscript (Line 77-85).

Comment 3: Can the results from this study be extrapolated to shunts from other brands? Please discuss.

Response We deeply appreciate the reviewer's insightful comment. This study conducted a rigorous clinical evaluation by comparing the target device with a control device that met equivalent safety and efficacy benchmarks. Our methodology adheres to the harmonized regulatory frameworks of both the FDA and China NMPA, specifically addressing devices with identical intended use and comparable technological characteristics. The preclinical validation demonstrated equivalence in critical performance endpoints, suggesting these findings may hold broader regulatory relevance for similarly classified devices [15]. However, brand-specific variations in material composition or design specifications may require additional verification when considering other manufacturers' products. We have incorporated this point into the revised Discussion section in the revised manuscript. And we are deeply grateful for the reviewer’s invaluable comment which has greatly improved the quality of our manuscript (Line 397-405).

---

## [Decision Letter · Decision Letter 1]

23 Jul 2025

Equivalence and Endothelial Safety of Temporary Intracoronary Shunt Device: Insights from Preclinical Porcine and Rabbit Models

PONE-D-25-20869R1

Dear Dr. Jiang,

We’re pleased to inform you that your manuscript has been judged scientifically suitable for publication and will be formally accepted for publication once it meets all outstanding technical requirements.

Kind regards,

Dr Redoy Ranjan, MBBS, MRCSEd, Ch.M., MS (CV&TS), FACS

Academic Editor

PLOS ONE

Additional Editor Comments (optional):

Review Comments to the Author

Reviewer #1: Hello

This is a good article

Many studies have been conducted in this field so far, and this article can complement previous articles and increase information in this field.

Good luck.

Reviewer #2: I review the manuscript and the surgical technique was described well. The results were shown in a scientific way, easy to read and to understand. The discussion section was great. The revised version is much better

Reviewer #3: No further comments. All my comments were replied and my suggestions accurately addressed. Congratulations on an excellent manuscript.

---

## [Editor Report · Acceptance letter]

PONE-D-25-20869R1

PLOS ONE

Dear Dr. Jiang,

I'm pleased to inform you that your manuscript has been deemed suitable for publication in PLOS ONE. Congratulations! Your manuscript is now being handed over to our production team.

Kind regards,

on behalf of

Dr. Redoy Ranjan

Academic Editor

PLOS ONE